# Quality of Oral Anticoagulation Control with Warfarin According to Sex: A Cross-Sectional Study

**DOI:** 10.3390/ijerph22010065

**Published:** 2025-01-06

**Authors:** Catiane Costa Viana, Marcus Fernando da Silva Praxedes, Mauro Henrique Nogueira Guimarães de Abreu, Waleska Jaclyn Freitas Nunes de Sousa, Cássia Rodrigues Lima Ferreira, Emílio Itamar de Freitas Campos, José Luiz Padilha da Silva, Maria Auxiliadora Parreiras Martins

**Affiliations:** 1Faculdade de Farmácia, Universidade Federal de Minas Gerais, Belo Horizonte 31270-901, Minas Gerais, Brazil; catianeviana@gmail.com (C.C.V.);; 2Centro de Ciências da Saúde, Universidade Federal do Recôncavo da Bahia, Santo Antônio de Jesus 44430-622, Bahia, Brazil; 3Faculdade de Odontologia, Universidade Federal de Minas Gerais, Belo Horizonte 31270-901, Minas Gerais, Brazil; 4Faculdade de Medicina, Universidade Federal de Minas Gerais, Belo Horizonte 30130-100, Minas Gerais, Brazil; 5Hospital das Clínicas, Universidade Federal de Minas Gerais, Belo Horizonte 30130-100, Minas Gerais, Brazil; 6Departamento de Estatística, Universidade Federal do Paraná, Curitiba 81531-980, Paraná, Brazil; 7Hospital Risoleta Tolentino Neves, Belo Horizonte 31744-012, Minas Gerais, Brazil

**Keywords:** outpatient care, cardiovascular diseases, oral anticoagulants, sex, thromboembolism, warfarin

## Abstract

Evidence indicates a difference between men and women in oral anticoagulation control, but the results were discrepant. This study investigated the association of sex with oral anticoagulation control in patients on warfarin assisted by anticoagulation clinics (ACs) in Brazil. The cross-sectional study included patients on warfarin recruited at three public ACs in southeast Brazil (2014–2015). The quality of oral anticoagulation was estimated by the time in therapeutic range (TTR). Univariable and multivariable linear regression models were built to examine the association of sociodemographic, behavior, clinical, and drug therapy variables with TTR. Overall, 801 participants were studied (455; 56.8% women), with a mean age of 65.0 (13.4) years. The female sex was associated with lower TTR than the male sex (Beta (95% CI) = −17.01 (−30.25; −3.76), *p* = 0.012), however, this difference decreased with increasing age, becoming null after age 60. Smoking patients had a lower TTR than non-smokers (−5.18 (−10.02; −0.34), *p* = 0.036). The results showed that the association of sex with oral anticoagulation control is dependent on age. Women have lower TTR than men, but this difference is null in older patients. Knowledge of these factors may be useful for developing strategies to improve care for these patients.

## 1. Introduction

Oral anticoagulants (OACs) are recommended for patients with cardiovascular diseases (CVDs) and risk factors for thromboembolism [1,2]. Direct oral anticoagulants (DOACs), such as direct factor Xa inhibitors and direct thrombin inhibitors, do not need constant laboratory monitoring. However, safety data in certain populations are still limited, including patients with mechanical heart valves, atrial fibrillation (AF) with rheumatic heart disease and mitral stenosis, and severe renal failure [3]. Additionally, they are more expensive and less accessible, especially in low- and middle-income countries. These limitations make vitamin K inhibitors (e.g., warfarin) the OACs of choice in these countries [1].

Despite its importance, warfarin therapy presents a wide variability in dose–response and risk of adverse events, requiring frequent laboratory monitoring to guide dose adjustments. The assessment of the quality of oral anticoagulation with warfarin is commonly estimated by time in the therapeutic range (TTR). The TTR determined by the Rosendaal method is based on a linear interpolation using at least two international normalized ratio (INR) values with a range of 0–100% [4]. Values of TTR < 65% indicate poor control and correlation with a higher risk of clinical complications, such as stroke, bleeding, and death [5].

The factors that may affect oral anticoagulation control are related to intrinsic characteristics of warfarin, such as variability in dose–response, interaction with other drugs [6,7], and diet (such as green leafy vegetables) [8,9]. Demographic factors (age [6,10], and sex [7]); social factors (level of patient knowledge [1] and access to services [11]), and behavioral factors (alcohol use [12] and smoking [6]) may also affect oral anticoagulation control. Moreover, clinical factors such as body mass index (BMI) [6] and comorbidities (e.g., renal failure and diabetes mellitus) [6,12], genetic polymorphisms [7], and the strategy of patient care (e.g., face-to-face or remote management) [13] were also associated with anticoagulation control with warfarin.

Previous studies reported differences between men and women in oral anticoagulation control, although with discrepant results. The comparison of sexes showed worse anticoagulation control in women than men [14,15,16] or no significant difference [10,12,17]. Sex-related pharmacokinetic (PK) and pharmacodynamic (PD) properties may also explain drug response. In addition, drug–food and drug–drug interactions influence factors of drug effect [18].

Regarding clinical conditions, women have been reported to present worse outcomes in CVDs than men, and this may be related to pathophysiologic and behavioral differences. There is evidence that women are undertreated for primary and secondary thromboprophylaxis compared to men [19]. Women with non-valvular AF on DOACs presented an overall relative risk of stroke and systemic embolism higher than men. Lower effectiveness in women was also observed in warfarin therapy [20]. Despite the higher risk of thromboembolic complications in women, underprescription of oral anticoagulation in AF women was reported, but literature results are conflicting [21,22]. Women were also reported to receive underdoses of DOACs more frequently than men [23].

Understanding the role of sex in TTR may be useful for establishing personalized strategies for patient care. This topic is crucial for pharmacists and other healthcare professionals involved in developing protocols tailored to patients’ specific needs. Most studies reporting the impact of sex on TTR were not designed specifically for this purpose, as highlighted in a previous systematic review with meta-analysis [14].

Likewise, sex and oral anticoagulation control have not been extensively studied in populations living low- and middle-income countries [24]. In this context, this topic should be better elucidated in patients living in Brazil, which is a continental country with singularities. The Brazilian Unified Health System is planned to provide full access to the population, but there are still several barriers to this access, such as racial, spatial, and income inequalities [25]. This study aimed to investigate the association of sex with oral anticoagulation control in patients on warfarin assisted by public anticoagulation clinics (ACs) in southeast Brazil.

## 2. Materials and Methods

### 2.1. Study Design and Participants

This is a cross-sectional study developed in three ACs located in Belo Horizonte, a southeastern city of Brazil. These are multidisciplinary services that play a referencing role in the Brazilian Public Health System, involving healthcare teams with pharmacists, physicians, and nurses. This study was conducted in accordance with the Declaration of Helsinki and approved by the Research Ethics Committee of the Universidade Federal de Minas Gerais (CAAE 08136613.4.0000.5149; date of approval: 18 December 2013). Informed consent was obtained from all subjects involved in the study.

The study was conducted in a sample composed of all patients recruited consecutively from 1 October 2014 to 31 December 2015, according to the inclusion and exclusion criteria. Inclusion criteria were: adult patients (≥18 years), both sexes, definitive diagnosis of heart disease, and presence of at least one indication for continuous use of at least two months of warfarin (e.g., AF, mechanical heart valve, history of deep vein thrombosis, pulmonary thromboembolism, intracardiac thrombus, and stroke). Patients expected to be using warfarin for less than 12 months, with a life expectancy of less than one year, poor cognitive status, and lack of an engaged caregiver were excluded.

### 2.2. Data Collection

Initially, sociodemographic, behavioral, clinical, and drug therapy data were collected by patients’ interviews and medical records. Complementary information was obtained from medical records and prescriptions. Sociodemographic data included age (years), sex, degree of schooling, and monthly income per capita (in American dollars (USD 1 = BRL 5.14)). Behavior data included history of alcohol consumption (defined as daily consumption of more than 60 g of ethanol) [26] and smoking (defined as the use of at least one cigarette in the last month) [27]. Clinical and drug therapy data included indication for oral anticoagulation, number and types of comorbidities defined as any diagnosis registered on medical records, number and type of drugs in chronic use taken for more than 30 days, including warfarin, weekly warfarin dose (milligrams), assistance for warfarin administration, and follow-up time at the AC (years).

The INR results were retrieved from the hospital’s computerized databases and used to calculate TTR, applying the Rosendaal method [4] adjusted for the target INR (2.00–3.00 or 2.50–3.50). Data were collected from 1 October 2014 to 31 December 2015, considering the maximum follow-up period for each patient. The database was built in Epi-Data software, version 3.1 (EpiData Assoc, Odense, Denmark) and validated by double entry.

### 2.3. Statistical Analysis

The variables used in the analysis were TTR, age, sex, degree of schooling, monthly income per capita (in American dollars (USD 1 = BRL 5.14)), history of alcohol consumption, smoking, indication for oral anticoagulation, number and type of comorbidities, number and type of drugs used for more than 30 days, including warfarin, weekly warfarin dose, assistance for warfarin administration, follow-up time at the AC, and target INR of 2.00–3.00 or 2.50–3.50. The selection of variables was based on previous studies [7,16,28].

Data were summarized using absolute and relative frequencies for categorical variables, as well as the mean and standard deviation for continuous variables. Data were described by sex and compared using likelihood ratio tests from linear generalized models. Univariable linear regression models were built for TTR using the following variables: (a) sociodemographic: age, sex, degree of schooling, monthly income per capita; (b) behavior variables: history of alcohol consumption and smoking; (c) clinical and drug therapy: indication for oral anticoagulation, comorbidities, drugs used for more than 30 days including warfarin, weekly warfarin dose, assistance for warfarin administration, follow-up time at the AC and target INR of 2.00–3.00 or 2.50–3.50. Monthly income per capita was log-transformed due to its asymmetric nature. Multivariable linear regression models considering TTR as a response variable were built by testing the explanatory variables described above. Unadjusted and adjusted Beta (95% confidence interval—95% CI) for all covariates were estimated. The selection of variables was performed manually by backward elimination and the final model included variables statistically significant at *p* < 0.05, with an estimation of Beta (95% CI) for those covariates. Interaction terms between the explanatory variables were considered during the modeling. An interaction plot was developed to depict the relationship between TTR and sex according to age. Non-linearities between TTR and continuous predictors were explored using natural cubic spline functions. Distributional assumptions were assessed using residual plots and normality tests. Due to a slight deviation from normality detected in the residual analysis, a linear model with robust standard errors was fitted as a sensitivity analysis. The main conclusion remained unchanged.

The significance level was set at 0.05 and all hypothesis tests were two-sided. All analyses were performed in the free R software, version 4.2.2, using the packages readxl, tidyverse, and ggplot2.

## 3. Results

Of the 801 eligible patients (455; 56.8% women), 312 (39.0%) were recruited from AC1, 242 (30.2%) from AC2, and 247 (30.8%) from AC3. The mean age of participants was 65.0 (13.4) years. The majority of patients presented elementary school (586; 73.2%) as the level of education, and the percentage of women with no school education (67; 14.7%) was higher than that of men (28; 8.1%). The mean monthly income per capita was USD 163.8 (USD 120.2), with women having lower monthly income per capita (USD 137.4 (USD 84.3)) than men (USD 198.5 (USD 148.4)). Among the alcohol or tobacco users, the majority were men (73.6% and 56.0%, respectively). Patients’ characteristics are detailed in Table 1.

Regarding comorbidities, the greatest differences observed between men and women were in congestive heart failure (men: 148; 42.8%, women: 164; 36.0%), valvular diseases (men: 81; 23.4%, women: 140; 30.8%), neuropsychiatric disorders (men: 31; 9.0%, women: 73; 16.0%), and arterial coronary disease (men: 23; 6.6%, women: 10; 2.2%). The mean number of drugs in chronic use, including warfarin, was 5.9 (2.2) and the greatest difference between the sexes was in the use of simvastatin (men: 188; 54.3%, women: 192; 42.2%).

The main indication for warfarin use was AF/flutter (640; 79.9%), followed by the use of mechanical heart valves (166; 20.7%). The majority of patients with mechanical heart valves were female (110; 66.3%). The mean weekly warfarin dose was 29.0 (14.6) mg with a predominance of target INR of 2.00–3.00 (661; 82.5%). The mean TTR was 62.6% (19.0%) (men: 63.7% (18.4%); women: 61.7% (19.6%)).

In the univariable linear regression, monthly income per capita was directly related to TTR (Beta (95% CI) = 1.01 (0.01; 2.01), *p* = 0.049), and smoking patients had lower TTR than non-smokers (Beta (95% CI) = −5.27 (−10.07; −0.47), *p* = 0.032). The final multivariable linear regression revealed that the female sex was associated with lower TTR than the male sex (Beta (95% CI) = −17.01 (−30.25; −3.76), *p* = 0.012). Smoking patients continued with lower TTR than non-smokers (−5.18 (−10.02; −0.34), *p* = 0.036). The results of these analyses are detailed in Table 2.

The interaction between age and sex was statistically significant, indicating that the association of sex with TTR was age-dependent. For each increase of one year in age, there was a 0.23 difference in TTR between sexes (Age * Female: Beta (95% CI) = 0.23 (0.03; 0.43), *p* = 0.026)). Since this difference was negative (initially equal to −17.01%), with increasing age, it decreased, becoming null after 60 years (Table 2). The interaction plot indicating the relationship between TTR and sex according to age is depicted in Figure 1.

The Kolmogorov–Smirnov normality test showed some deviation (*p* = 0.012). In the sensitivity analysis (model adjusted with robust standard errors) the inferences were similar. The *p*-value for the interaction Age * Sex went from *p* = 0.026 to *p* = 0.028, confirming the tendency for TTR to differ between men and women according to age. The *p*-value for the association of smoking with TTR went from *p* = 0.036 to *p* = 0.055, very close to significance at the 5% level.

## 4. Discussion

This study assessed the association of sex with oral anticoagulation control with warfarin in patients assisted by public ACs in Brazil. Our findings revealed that female sex was associated with lower TTR compared to male sex (Beta (95% CI) = −17.01 (−30.25; −3.76), *p* = 0.012). Measuring the clinical relevance of this difference is challenging, since the relationship between variations in TTR and the risk of complications is unclear in the literature. A study showed that a 10% increase in TTR correlated with a −0.32%/patient-year decrease in stroke/systemic embolism rate [29]. However, it has not been assessed whether the increases in these risks may differ according to sex.

In previous studies, women also presented worse oral anticoagulation control, however, the reasons for these differences were not clear [14,15,30]. There is a lack of studies designed specifically to investigate the association of sex with oral anticoagulation control with warfarin [14,15]. The female sex negatively affects anticoagulation control according to the SAMe-TT2R2 score (female sex, age < 60 years, medical history (>2 comorbidities), treatment (interacting drugs, e.g., amiodarone for rhythm control), tobacco use (doubled), race (doubled)), developed for predicting poor INR control in AF patients treated with vitamin K inhibitors [31].

The investigation of sex-related differences in oral anticoagulation control is important to understand their role in the worse results attributed to women with CVDs [19,32]. The main indication for warfarin use in this study was AF with women presenting a lower proportion of this diagnosis. Previous data reported that AF women experienced worse symptoms and quality of life and a higher risk of stroke and death [33,34]. Women tend to be undertreated with oral anticoagulation and female sex is an independent risk factor for thromboembolism in AF [34].

Lower TTR in women may be explained in part by sociodemographic and behavioral factors. In this study, women had lower education levels and income than men. Lower education and lower income were associated with lower health literacy [35,36]. This may result in limited ability of self-management of drug therapy and reduced autonomy for patients to understand and follow healthcare recommendations [7]. Thus, women may be less engaged with the shared decision-making process [37]. Additionally, a lower economic status may hinder access to medications and healthcare services and impact the medication adherence and outcomes in oral anticoagulation [34,38]. Therefore, targeted care should be developed to assist women given their lower mean incomes, lack of basic healthcare, and disproportional impact of chronic diseases [39].

Women’s access to healthcare and medication adherence are marked by socially and culturally constructed barriers that affect all stages of care, resulting in suboptimal treatments and outcomes [40]. Women from disadvantaged backgrounds are more likely to experience inequalities along the way, due to intersecting factors, such as ethnicity, race, and poverty [41,42]. They also frequently face biases due to gender stereotypes when receiving healthcare [41]. Issues related to beliefs, religion, and lack of support from family and community are also barriers that hinder women’s access to healthcare [43]. In addition to these factors, women are underrepresented in clinical trials, reflected in treatment inequalities between sexes [44]. Although little evaluated, it is possible these barriers lead to sex differences in TTR results.

Our findings revealed that smoking is associated with lower TTR. Previous findings showed that current smokers presented a significantly higher risk of poor anticoagulation control compared to non-smokers [6]. The reasons for this difference remain unclear. Possible explanations involve the potential of smoking to increase warfarin clearance and to reduce anticoagulant effects [45]. These aspects deserve further studies to clarify whether this impact differs between sexes. Women were reported to present higher risk of clinical complications than men due to smoking (e.g., death at younger ages and stroke) [46]. In this study, women had a lower frequency of smoking habits and alcohol consumption. Alcohol consumption may also impact TTR [12] and increase the risk of major bleeding in warfarin therapy. Differences in alcohol metabolism between sexes are complex [47] and the impact of alcohol on TTR between sexes has not been described so far.

Comorbidities have also been associated with worse TTR [12], but the mechanisms were not detailed in depth. Physiological changes, increased number of medications taken, and medication non-adherence seem to be involved in this aspect [48,49]. Physiological characteristics should be considered when assessing anticoagulation differences, as well as transitions in disease presentation between sexes over time. Comorbidities were reported to negatively affect TTR in both sexes, but no findings clearly explained the lower TTR in women [30]. Finally, there was no significant difference between sexes in this study regarding the number of drugs in chronic use.

We demonstrated that the association of sex with oral anticoagulation control is dependent on age. At ages < 60 years, there is a considerable difference in TTR between sexes with women presenting lower TTR. However, with increasing age, this difference decreases, becoming null in older patients. We did not find similar results in the literature. There are social, cultural, and behavioral factors that may negatively affect anticoagulation control in women aged < 60 years compared to older women. In Brazil, as in other countries, women spend more time in unpaid work, associated with household activities and family care. This disparity may have a greater impact on women’s health [50,51]. These factors may influence their adherence to oral anticoagulation, although we did not find evidence to evaluate this hypothesis. Furthermore, it is possible that hormonal differences between women aged < 60 years and women aged ≥ 60 years, or the use of hormonal contraceptives by women of childbearing age, may explain our results but deserve more clarifications.

Based on our results and the process of population aging, there is a need for care practices adjusted to older women to promote self-care and improve health outcomes. Since women have a longer life expectancy than men, a larger population of older women is expected to use oral anticoagulants. Protocols for care practice and management of oral anticoagulation should include health education strategies to improve health literacy, considering the vulnerabilities of this group of patients [14,33,37].

The strength of our study was the evaluation of the association of sex with oral anticoagulation control in patients from Brazil, a country marked by its social and cultural diversity. It is estimated that about 80% of CVD-related deaths occur in low- and middle-income countries, regions where these diseases and the burden of risk factors tend to increase as a result of an ongoing epidemiological transition [52]. Our findings support the implementation of more effective interventions focused on the reduction of warfarin-related risks between sexes improving healthcare delivery and access to more qualified healthcare.

There are limitations to be addressed. This paper is a post hoc analysis of a previous cross-sectional study from 10 years ago, when the use of DOACs was not so prevalent. The study design is cross-sectional which hinders causality assessment. The development of longitudinal studies on the topic could be useful to confirm these associations over time. Furthermore, relevant data for comparison between sexes could not be obtained, such as hormonal factors, genetic polymorphisms (e.g., CYP2C9/VKORC1), and diet (vitamin K intake). The lack of these data that could influence warfarin metabolism limits the depth of the analysis. Data collection involved patient interviews which may imply memory bias. Finally, this study was developed in a single region of Brazil, and the generalizability of our findings across Brazilian states or to other South American countries remains unknown.

## 5. Conclusions

The results showed that the association of sex with oral anticoagulation control is dependent on age. Women had lower TTR than men, but this difference decreased with age, becoming null in older patients. Smoking was associated with lower TTR. We also discussed the challenges of addressing the role of sex in oral anticoagulation control, given that many factors related or unrelated to sex may affect TTR and are still unclear in the literature. Our findings support the implementation of interventions focused on the reduction of warfarin-related risks between sexes by improving access to more qualified healthcare.

## Figures and Tables

**Figure 1 ijerph-22-00065-f001:**
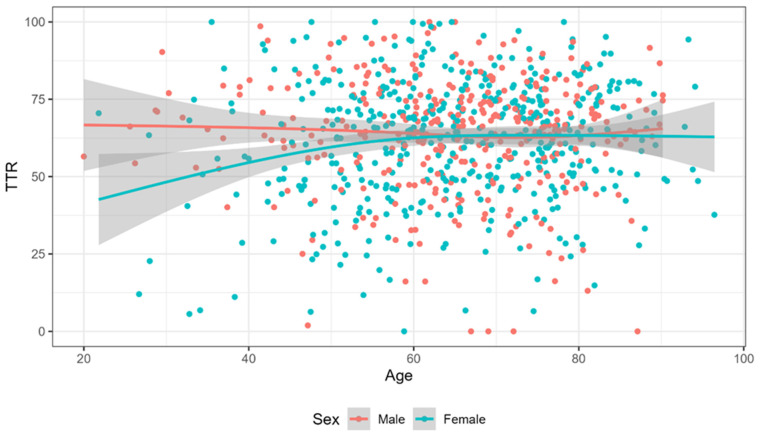
Relationship between TTR and sex by age.

**Table 1 ijerph-22-00065-t001:** Description of the characteristics of the total population and stratified by sex of patients on warfarin.

	Total	Men	Women	*p*-Value
*n* (%)	801 (100)	346 (43.2)	455 (56.8)	
Age, mean (SD)	65 (13.4)	65.3 (13.1)	64.8 (13.6)	0.658
Degree of schooling, *n* (%)				
No school education	95 (11.9)	28 (8.1)	67 (14.7)	0.002
Elementary school	586 (73.2)	254 (73.4)	332 (73.0)	
High school	104 (13.0)	53 (15.3)	51 (11.2)	
Undergraduate degree	14 (1.7)	10 (2.9)	4 (0.9)	
No information	2 (0.2)	1 (0.3)	1 (0.2)	
Monthly income per capita—USD, mean (SD)	163.8 (120.2)	198.5 (148.4)	137.4 (84.3)	<0.001
History of alcohol consumption, *n* (%)	72 (9.0)	53 (15.3)	19 (4.2)	<0.001
Smoking, *n* (%)	66 (8.2)	37 (10.7)	29 (6.4)	0.029
Indication for anticoagulant therapy, *n* (%)				
AF/Flutter	640 (79.9)	278 (80.3)	362 (79.6)	0.783
General mechanical prosthesis	166 (20.7)	56 (16.2)	110 (24.2)	0.005
Ischemic stroke and/or transient ischemic attack	148 (18.5)	75 (21.7)	73 (16.0)	0.043
Deep vein thrombosis and/or pulmonary thromboembolism	27 (3.4)	10 (2.9)	17 (3.7)	0.508
Comorbidities, mean (SD)	3.4 (1.9)	3.3 (1.8)	3.4 (1.9)	0.373
Comorbidities (%)				
Systemic arterial hypertension	609 (76.0)	267 (77.2)	342 (75.2)	0.510
Congestive heart failure	312 (39.0)	148 (42.8)	164 (36.0)	0.053
Dyslipidemia	277 (34.6)	130 (37.6)	147 (32.3)	0.121
Valvular diseases	221 (27.6)	81 (23.4)	140 (30.8)	0.020
Diabetes	180 (22.5)	80 (23.1)	100 (22.0)	0.701
Respiratory diseases	176 (22.0)	66 (19.1)	110 (24.2)	0.083
Neuropsychiatric disorders	104 (13.0)	31 (9.0)	73 (16.0)	0.003
Chagas disease	88 (11.0)	34 (9.8)	54 (11.9)	0.358
Endocardium, myocardium, pericardium diseases	86 (10.7)	41 (11.8)	45 (9.9)	0.376
Aortic disease	78 (9.7)	29 (8.4)	49 (10.8)	0.256
Arrhythmias	70 (8.7)	34 (9.8)	36 (7.9)	0.344
Gastrointestinal tract diseases	47 (5.9)	16 (4.6)	31 (6.8)	0.187
Renal insufficiency	45 (5.6)	25 (7.2)	20 (4.4)	0.087
Arterial coronary disease	33 (4.1)	23 (6.6)	10 (2.2)	0.002
Peripheral vascular diseases	23 (2.9)	9 (2.6)	14 (3.1)	0.688
Neoplasms	19 (2.4)	11 (3.2)	8 (1.8)	0.193
Liver failure	10 (1.2)	3 (0.9)	7 (1.5)	0.388
Drugs in chronic use—including warfarin, mean (SD)	5.9 (2.2)	5.8 (2.2)	6.0 (2.2)	0.174
Drugs in chronic use—including warfarin, *n* (%)				
Simvastatin	380 (47.4)	188 (54.3)	192 (42.2)	0.001
Furosemide	366 (45.7)	151 (43.6)	215 (47.3)	0.309
Atenolol	183 (22.8)	69 (19.9)	114 (25.1)	0.086
Spironolactone	182 (22.7)	86 (24.9)	96 (21.1)	0.210
Acetylsalicylic acid	163 (20.3)	73 (21.1)	90 (19.8)	0.647
Omeprazole	155 (19.4)	52 (15.0)	103 (22.6)	0.006
Hydrochlorothiazide	147 (18.4)	51 (14.7)	96 (21.1)	0.020
Amlodipine	120 (15.0)	59 (17.1)	61 (13.4)	0.154
Propranolol	85 (10.6)	30 (8.7)	55 (12.1)	0.117
Amiodarone	70 (8.7)	33 (9.5)	37 (8.1)	0.487
Fluoxetine	42 (5.2)	10 (2.9)	32 (7.0)	0.007
Metoprolol	28 (3.5)	13 (3.8)	15 (3.3)	0.726
Carbamazepine	12 (1.5)	6 (1.7)	6 (1.3)	0.633
Phenytoin	8 (1.0)	4 (1.2)	4 (0.9)	0.698
Average weekly dose, mean (SD)	29.0 (14.6)	28.6 (14.8)	29.4 (14.5)	0.465
Assistance for warfarin administration, *n* (%)	150 (18.7)	69 (20.0)	81 (17.8)	0.443
Follow-up time at the anticoagulation clinic—years, mean (SD)	3.4 (3.3)	3.2 (3.0)	3.6 (3.5)	0.145
Target INR, *n* (%)				
2.0–3.0	661 (82.5)	304 (87.9)	357 (78.5)	<0.001
2.5–3.5	140 (17.5)	42 (12.1)	98 (21.5)	
TTR, mean (SD)	62.6 (19.0)	63.7 (18.4)	61.7 (19.6)	0.141

Abbreviation: AF, atrial fibrillation; INR, international normalized ratio; SD, standard deviation; TTR, time in therapeutic range; US, United States. USD 1 = BRL 5.14 (9 May 2024).

**Table 2 ijerph-22-00065-t002:** Characteristics associated with time in therapeutic range (TTR) in patients on warfarin: linear regression model.

Characteristics	Beta (95% CI), *p*-Value Unadjusted	Beta (95% CI), *p*-Value Multivariable Model	Beta (95% CI), *p*-Value Final Model
Age			
Age	0.08 (−0.02; 0.18), *p*= 0.117	−0.07 (−0.24; 0.09), *p* = 0.388	−0.07 (−0.22; 0.08), *p* = 0.375
Sex			
Male	(ref)	(ref)	(ref)
Female	−2.00 (−4.67; 0.67), *p* = 0.142	−16.77 (−30.26; −3.29), *p* = 0.015	−17.01 (−30.25; −3.76), *p* = 0.012
Interaction Age * Sex			
Age * Female	-	0.23 (0.02; 0.43), *p* = 0.029	0.23 (0.03; 0.43), *p* = 0.026
Degree of schooling			
No school education	(ref)	(ref)	
Elementary school	−3.55 (−7.69; 0.59), *p* = 0.093	−2.18 (−6.50; 2.13), *p* = 0.321	
High school	−1.77 (−7.09; 3.54), *p* = 0.514	−0.68 (−6.41; 5.06), *p* = 0.817	
Undergraduate degree	−4.99 (−15.71; 5.74), *p* = 0.362	−4.64 (−15.57; 6.30), *p* = 0.406	
Monthly income per capita—USD			
Monthly income per capita (in log)	1.01 (0.01; 2.01), *p* = 0.049	0.77 (−0.30; 1.84), *p* = 0.157	
History of alcohol consumption			
No	(ref)	(ref)	
Yes	−1.25 (−5.87; 3.38), *p* = 0.598	−1.47 (−6.27; 3.34), *p* = 0.550	
Smoking			
No	(ref)	(ref)	(ref)
Yes	−5.27 (−10.07; −0.47), *p* = 0.032	−4.27 (−9.23; 0.69), *p* = 0.092	−5.18 (−10.02; −0.34), *p* = 0.036
Indication for anticoagulant therapy			
General mechanical prosthesis			
No	(ref)	(ref)	
Yes	−0.58 (−3.84; 2.69), *p* = 0.730	1.54 (−4.66; 7.75), *p* = 0.629	
Ischemic stroke and/or transient ischemic attack			
No	(ref)	(ref)	
Yes	2.41 (−1.0; 5.81), *p* = 0.167	1.69 (−1.99; 5.37), *p* = 0.368	
Comorbidities			
Arterial coronary disease			
No	(ref)	(ref)	
Yes	3.00 (−3.66; 9.66), *p* = 0.377	2.16 (−4.63; 8.96), *p* = 0.533	
Neuropsychiatric disorders			
No	(ref)	(ref)	
Yes	2.41 (−1.53; 6.34), *p* = 0.231	2.86 (−1.18; 6.90), *p* = 0.166	
Drug in chronic use—including warfarin			
Simvastatin			
No	(ref)	(ref)	
Yes	2.31 (−0.33; 4.96), *p* = 0.087	1.31 (−1.51; 4.13), *p* = 0.363	
Target INR			
2.0–3.0	(ref)	(ref)	
2.5–3.5	−1.02 (−4.50; 2.47), *p* = 0.568	−0.19 (−6.79; 6.40), *p* = 0.954	

Abbreviation: CI, confidence interval; INR, international normalized ratio; SD, standard deviation; US, United States. USD 1 = BRL 5.14 (9 May 2024). * Significant *p* value (*p* < 0.05).

## Data Availability

The dataset used and analyzed during the current study is available from the corresponding author on reasonable request.

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
