# Peer review of "Quality of Oral Anticoagulation Control with Warfarin According to Sex: A Cross-Sectional Study"

_ijerph, 2025, doi:10.3390/ijerph22010065_

Round 1
Reviewer 1 Report
Comments and Suggestions for Authors
The presented manuscript has been done in a clear manner and provides relevant results. However, some minor remarks still remain. 1. Currently, the situation with oral anticoagulant therapy is changing very dynamically. In connection with this, the relevance of this study is somewhat reduced by the fact that the recruitment of patients was carried out a long time ago. Over this time direct oral anticoagulants (DOAC) have become the first-line therapy for non-valvular atrial fibrillation and venous thromboembolic disease. There remains a small niche for the use of VKA. The economic benefit of using the latter is not so obvious, given the need for constant, not free, laboratory monitoring. In this regard, it is necessary to point out, at least in the limitations, that this paper is a post hoc analysis of a previous cross-sectional study from 10 years ago, when the use of DOAC was not so prevalent. 2. I have some doubts regarding the correctness of the terminology. It is known that regression coefficient indicates the strength and direction of relationship between independent variables and dependent variable. When we talk about an effect, we mean a cause-and-effect relationship that can be established in a longitudinal, not a cross-sectional study. The authors clearly write about this when mentioning the limitations of the study (line 279): “The study design is cross-sectional which hinders the causality assessment”. While describing the relationship between independent variables and the dependent ones (i.e. TTR), the authors use the term “association”. But when it comes to sex, they use the term “effect”. In my opinion, the strength of the relationship is still not the same thing as the effect or influence. Therefore, the authors should make appropriate corrections or clearly justify their position regarding this terminology. 3. There is a typo in table 1: correct "valor" to "value".
Author Response
Author's Reply to the Review Report (Reviewer 1)
Response to Reviewer X Comments
- Summary
Thank you very much for taking the time to review this manuscript. We appreciate the careful revision and suggestions provided to improve the quality of our work. In the following text, my coauthors and I summarized your concerns followed by an itemized list of all changes made and our rebuttal to the comments. Changes in the manuscript are identified by red font.
- Point-by-point response to Comments and Suggestions for Authors
Comments 1: The presented manuscript has been done in a clear manner and provides relevant results. However, some minor remarks still remain.
- Currently, the situation with oral anticoagulant therapy is changing very dynamically. In connection with this, the relevance of this study is somewhat reduced by the fact that the recruitment of patients was carried out a long time ago. Over this time direct oral anticoagulants (DOAC) have become the first-line therapy for non-valvular atrial fibrillation and venous thromboembolic disease. There remains a small niche for the use of VKA. The economic benefit of using the latter is not so obvious, given the need for constant, not free, laboratory monitoring. In this regard, it is necessary to point out, at least in the limitations, that this paper is a post hoc analysis of a previous cross-sectional study from 10 years ago, when the use of DOAC was not so prevalent.
Response 1: Thank you for turning our attention to this topic. As suggested by the reviewer, we highlighted in the limitations that this article is a post hoc analysis of a previous cross-sectional study from 10 years ago, when the use of DOACs was not so prevalent.
Manuscript modifications – Discussion section, page 9, paragraph 11, lines 296 and 297
“This paper is a post hoc analysis of a previous cross-sectional study from 10 years ago, when the use of DOACs was not so prevalent.”
Comments 2:
- I have some doubts regarding the correctness of the terminology. It is known that regression coefficient indicates the strength and direction of relationship between independent variables and dependent variable. When we talk about an effect, we mean a cause-and-effect relationship that can be established in a longitudinal, not a cross-sectional study. The authors clearly write about this when mentioning the limitations of the study (line 279): “The study design is cross-sectional which hinders the causality assessment”. While describing the relationship between independent variables and the dependent ones (i.e. TTR), the authors use the term “association”. But when it comes to sex, they use the term “effect”. In my opinion, the strength of the relationship is still not the same thing as the effect or influence. Therefore, the authors should make appropriate corrections or clearly justify their position regarding this terminology.
Response 2: We agree with reviewer´s comments on the correctness of the terminologies "effect" and "association". We have revised the manuscript and added corrections to improve its conceptual consistency.
Manuscript modifications – Abstract, page 1, line 17
“This study investigated the association of sex with oral anticoagulation control in patients on warfarin assisted by anticoagulation clinics (ACs) in Brazil.”
Manuscript modifications – Abstract, page 1, lines 21 to 27
“(…) Univariable and multivariable linear regression models were built to examine the association of sociodemographic, behaviour, clinical, and drug therapy variables with TTR. Overall, 801 participants were studied (455; 56.8% women), with a mean age of 65.0 (13.4) years. The female sex was associated with lower TTR than the male sex [Beta (95% CI)=-17.01 (-30.25; -3.76), p=0.012], however, this difference decreased with increasing age, becoming null after age 60. Smoking patients had a lower TTR than non-smokers [-5.18 (-10.02; -0.34), p=0.036]. The results showed that the association of sex with oral anticoagulation control is dependent on age.”
Manuscript modifications – Introduction section, page 2, paragraph 6, lines 75 and 76
“Understanding the role of sex on TTR may be useful for establishing personalized strategies for patient care.”
Manuscript modifications – Introduction section, page 2, paragraph 7 and lines 85 to 87
“This study aimed to investigate the association of sex with oral anticoagulation control in patients on warfarin assisted by public anticoagulation clinics (ACs) in Southeast Brazil.”
Manuscript modifications – Results section, page 6, paragraph 5, lines 192 and 193
“The interaction between age and sex was statistically significant, indicating that the association of sex with TTR was age-dependent.”
Manuscript modifications – Results section, page 7, paragraph 6, lines 203 to 205
“The p-value for the association of smoking with TTR went from p=0.036 to p=0.055, very close to significance at the 5% level.”
Manuscript modifications – Discussion section, page 7, paragraph 1, lines 207 and 208
“This study assessed the association of sex with oral anticoagulation control with warfarin in patients assisted by public ACs in Brazil.”
Manuscript modifications – Discussion section, page 7, paragraph 2, lines 216 and 217
“There is a lack of studies designed specifically to investigate the association of sex with oral anticoagulation control with warfarin [14, 15].”
Manuscript modifications – Discussion section, page 8, paragraph 8, lines 269 and 270
“We demonstrated that the association of sex with oral anticoagulation control is dependent on age.”
Manuscript modifications – Discussion section, page 9, paragraph 10, lines 288 to 290
“The strength of our study was the evaluation of the association of sex with oral anticoagulation control in patients from Brazil, a country marked by its social and cultural diversity.”
Manuscript modifications – Conclusion section, page 9, lines 308 to 313
“The results showed that the association of sex with oral anticoagulation control is dependent of age. Women had lower TTR than men, but this difference decreased with age, becoming null in older patients. Smoking was associated with lower TTR. We also discussed the challenges of addressing the role of sex on oral anticoagulation control, given that many factors related or unrelated to sex may affect TTR and are still unclear in the literature.”
Comments 3:
- There is a typo in table 1: correct "valor" to "value".
Response 3: Thank you for your careful review. We have corrected “valor” to “value” in table 1.
Reviewer 2 Report
Comments and Suggestions for Authors
Positive Aspects
-
Clinical Relevance: This article addresses a very important issue by examining the differences in oral anticoagulation control with warfarin between men and women. This type of study can be really helpful in tailoring treatments to improve outcomes for patients, as it acknowledges that biological and social factors may influence each sex differently.
-
Appropriate Methodology: The use of a cross-sectional design makes sense here, and the application of the Rosendaal method to calculate time in therapeutic range (TTR) is a solid choice. Including sociodemographic, clinical, and behavioral variables is a strong point because it provides a more comprehensive view of the factors that may affect anticoagulation control.
-
Clear Results: The findings are pretty clear: there is an association between female sex and lower TTR, as well as a negative impact of smoking. The relationship between age and TTR is also explained well, making the results easy to interpret.
-
Relevant Discussion: The discussion is well-developed, and the results are effectively placed within the context of existing literature. The authors provide reasonable explanations for the observed differences, offering a good understanding of why these findings might occur.
Areas for Improvement
-
Deeper Analysis: While the results are interesting, I think the study could benefit from a deeper look into the cultural and social factors that affect women’s access to healthcare and adherence to treatment. A better understanding of these aspects could be key to improving adherence and treatment outcomes for women.
-
Methodological Limitations:
- The cross-sectional design limits the ability to establish causal relationships. It would be interesting to see longitudinal studies that help confirm these associations over time.
- The lack of data on genetic or dietary factors that could influence warfarin metabolism (such as CYP2C9 polymorphisms or vitamin K intake) limits the depth of the analysis.
-
Practical Implications: The article could provide more concrete suggestions for improving clinical management, especially for women. Addressing modifiable factors like smoking and offering clear recommendations on how to improve treatment adherence would be valuable for clinicians.
Final Recommendation
The article is relevant and well-structured, but I think it could be enriched by a deeper exploration of the barriers women face in managing anticoagulation. Additionally, including more practical recommendations would make the study’s results more applicable to clinical practice. With these minor adjustments, it would be a very valuable contribution to the field.
Author Response
Author's Reply to the Review Report (Reviewer 2)
Response to Reviewer X Comments
- Summary
Thank you very much for taking the time to review this manuscript. We appreciate the careful revision and suggestions provided to improve the quality of our work. In the following text, my coauthors and I summarized your concerns followed by an itemized list of all changes made and our rebuttal to the comments. We have revised the manuscript text and made significant changes to the discussion section. Changes in the manuscript are identified by red font.
- Point-by-point response to Comments and Suggestions for Authors
Positive Aspects
- Clinical Relevance: This article addresses a very important issue by examining the differences in oral anticoagulation control with warfarin between men and women. This type of study can be really helpful in tailoring treatments to improve outcomes for patients, as it acknowledges that biological and social factors may influence each sex differently.
- Appropriate Methodology: The use of a cross-sectional design makes sense here, and the application of the Rosendaal method to calculate time in therapeutic range (TTR) is a solid choice. Including sociodemographic, clinical, and behavioral variables is a strong point because it provides a more comprehensive view of the factors that may affect anticoagulation control.
- Clear Results: The findings are pretty clear: there is an association between female sex and lower TTR, as well as a negative impact of smoking. The relationship between age and TTR is also explained well, making the results easy to interpret.
- Relevant Discussion: The discussion is well-developed, and the results are effectively placed within the context of existing literature. The authors provide reasonable explanations for the observed differences, offering a good understanding of why these findings might occur.
Comments 1:
Deeper Analysis: While the results are interesting, I think the study could benefit from a deeper look into the cultural and social factors that affect women’s access to healthcare and adherence to treatment. A better understanding of these aspects could be key to improving adherence and treatment outcomes for women.
Response 1: Thank you for this comment. We take a deeper approach about the cultural and social factors that affect women’s access to health care and adherence to treatment.
Manuscript modifications – Discussion section, page 8, paragraph 5, lines 240 to 249
“Women’s access to healthcare and medication adherence is marked by socially and culturally constructed barriers that affect all stages of care, resulting in suboptimal treatments and outcomes [40]. Women from disadvantaged backgrounds are more likely to experience inequalities along the way, due to intersecting factors, such as ethnicity, race and poverty [41, 42]. They also frequently face biases due to gender stereotypes when receiving healthcare [41]. Issues related to beliefs, religion, and lack of support from family and community are also barriers that hinder women’s access to healthcare [43]. In addition to these factors, women are underrepresented in clinical trials, reflecting in treatment inequalities between sexes [44]. Although little evaluated, it is possible these barriers lead to sex-differences in TTR results.”
Comments 2:
Methodological Limitations:
The cross-sectional design limits the ability to establish causal relationships. It would be interesting to see longitudinal studies that help confirm these associations over time.
The lack of data on genetic or dietary factors that could influence warfarin metabolism (such as CYP2C9 polymorphisms or vitamin K intake) limits the depth of the analysis.
Response 2: Thank you for your comment. We added a comment on the topic in the study limitations.
Manuscript modifications – Discussion section, page 9, paragraph 11 and lines 296 to 303
“There are limitations to be addressed. This paper is a post hoc analysis of a previous cross-sectional study from 10 years ago, when the use of DOACs was not so prevalent. The study design is cross-sectional which hinders the causality assessment. The development of longitudinal studies on the topic could be useful to confirm these associations over time. Furthermore, relevant data for comparison between sexes could not be obtained, such as hormonal factors, genetic polymorphisms (e.g., CYP2C9/VKORC1) and diet (vitamin K intake). The lack of these data that could influence warfarin metabolism limits the depth of the analysis. (…)”
Comments 3:
Practical Implications: The article could provide more concrete suggestions for improving clinical management, especially for women. Addressing modifiable factors like smoking and offering clear recommendations on how to improve treatment adherence would be valuable for clinicians.
Response 3: We are grateful with this suggested. We added more concrete suggestions to improve clinical management, especially for women.
Manuscript modifications – Discussion section, page 9, paragraph 9 and lines 282 to 287
“Based on our results and the process of population aging, there is a need for care practices adjusted to older women, to promote self-care and improve health outcomes. Since women have a longer life expectancy than men, a larger population of older women is expected to use oral anticoagulants. Protocols for care practice and management of oral anticoagulation should include health education strategies to improve health literacy, considering the vulnerabilities of this group of patients [14, 33, 37].”
Final Recommendation
The article is relevant and well-structured, but I think it could be enriched by a deeper exploration of the barriers women face in managing anticoagulation. Additionally, including more practical recommendations would make the study’s results more applicable to clinical practice. With these minor adjustments, it would be a very valuable contribution to the field.
Response: We appreciate the reviewer's careful review. We added considerations to the manuscript regarding the barriers women face in accessing health care and managing anticoagulation. We have also included more practical recommendations to improve the management of anticoagulant therapy.
Manuscript modifications – Discussion section, page 8, paragraph 5, lines 240 to 249
“Women’s access to healthcare and medication adherence is marked by socially and culturally constructed barriers that affect all stages of care, resulting in suboptimal treatments and outcomes [40]. Women from disadvantaged backgrounds are more likely to experience inequalities along the way, due to intersecting factors, such as ethnicity, race and poverty [41, 42]. They also frequently face biases due to gender stereotypes when receiving healthcare [41]. Issues related to beliefs, religion, and lack of support from family and community are also barriers that hinder women’s access to healthcare [43]. In addition to these factors, women are underrepresented in clinical trials, reflecting in treatment inequalities between sexes [44]. Although little evaluated, it is possible these barriers lead to sex-differences in TTR results.”
Manuscript modifications – Discussion section, page 9, paragraph 9 and lines 282 to 287
“Based on our results and the process of population aging, there is a need for care practices adjusted to older women, to promote self-care and improve health outcomes. Since women have a longer life expectancy than men, a larger population of older women is expected to use oral anticoagulants. Protocols for care practice and management of oral anticoagulation should include health education strategies to improve health literacy, considering the vulnerabilities of this group of patients [14, 33, 37].”